# The Effectiveness of Virtual Reality Exercise on Individual’s Physiological, Psychological and Rehabilitative Outcomes: A Systematic Review

**DOI:** 10.3390/ijerph17114133

**Published:** 2020-06-10

**Authors:** Jiali Qian, Daniel J. McDonough, Zan Gao

**Affiliations:** 1School of Kinesiology, the University of Minnesota, 1900 University Ave. SE, Minneapolis, MN 55455, USA; qian0133@umn.edu (J.Q.); mcdo0785@umn.edu (D.J.M.); 2Department of Physical Education, Shanghai Jiao Tong University, Minhang District, Shanghai 200240, China

**Keywords:** health promotion, immersive virtual reality, non-immersive virtual reality, interactive virtual reality, physical activity

## Abstract

*Objective purpose:* This review synthesized the literature examining the effects of virtual reality (VR)-based exercise on physiological, psychological, and rehabilitative outcomes in various populations. *Design:* A systematic review. *Data sources:* 246 articles were retrieved using key words, such as “VR”, “exercise intervention”, “physiological”, “psychology”, and “rehabilitation” through nine databases including Academic Search Premier and PubMed. *Eligibility criteria for selecting studies:* 15 articles which met the following criteria were included in the review: (1) peer-reviewed; (2) published in English; (3) randomized controlled trials (RCTs), controlled trials or causal-comparative design; (4) interventions using VR devices; and (5) examined effects on physiological, psychological, and/or rehabilitative outcomes. Descriptive and thematic analyses were used. *Results:* Of the 12 articles examining physiological outcomes, eight showed a positive effect on physical fitness, muscle strength, balance, and extremity function. Only four articles examined the effects on psychological outcomes, three showed positive effects such that VR exercise could ease fatigue, tension, and depression and induce calmness and enhance quality of life. Nine articles investigated the effects of VR-based exercise on rehabilitative outcomes with physiological and/or psychological outcomes, and six observed significant positive changes. In detail, patients who suffered from chronic stroke, hemodialysis, spinal-cord injury, cerebral palsy in early ages, and cognitive decline usually saw better improvements using VR-based exercise. *Conclusion:* The findings suggest that VR exercise has the potential to exert a positive impact on individual’s physiological, psychological, and rehabilitative outcomes compared with traditional exercise. However, the quality, quantity, and sample size of existing studies are far from ideal. Therefore, more rigorous studies are needed to confirm the observed positive effects.

## 1. Introduction

Over the past decades, the effects of physical activity (PA) on individual’s health have been well documented [1,2,3]. However, despite the well-known benefits of PA participation, according to the World Health Organization (WHO) approximately 25% of adults and 80% of adolescents around the world are physically inactive partly due to societal and lifestyle changes [4]. Exercise (i.e., planned, structured and repetitive PA) is often perceived as boring and hard, thereby causing adults and students shy away from PA-related behaviors after long days of work and/or school. Instead, individuals are more interested in leisure activities, such as video games, where entertainment can be obtained while relaxing (i.e., sedentary behavior). Thus, the combination of video games and engaging in PA (e.g., virtual reality (VR)-integrated exercise) may trigger their interest and improve their PA behavior.

In recent years, VR exercise has been recognized as a new approach to promote PA and health behaviors [5] and is becoming increasingly used in health promotion. Researchers have observed VR exercise to enhance the psychological benefits of exercise and increase the likelihood of long-term adherence to exercise [6,7]. VR is operationally defined as digital technology wherein sensory experiences, (e.g., visual, auditory, touch, and scent stimuli) are artificially created, prompting users to manipulate the objects within virtual environment [8]. In general, there are three types of VR: immersive, non-immersive, and interactive. Immersive VR utilizes head-mounted displays, body movement sensors, real-time graphics, and advanced interface devices (e.g., dedicated headsets) to simulate a completely virtual environment for users, whereas non-immersive VR utilizes an interface, such as a flat screen TV/computer screen, and requires the use of a corresponding keyboard, controller and/or joystick [9,10]. Interactive VR is centered on the user’s ability to interact with virtual objects through devices (e.g., gloves, digital glasses) which produce the sensation of manipulating real items, such as picking up an apple [11].

The development of VR technology and its utility during PA via its integration with traditional exercise equipment and rehabilitation practices has attracted attention in the fields of kinesiology and public health. As a therapeutic tool, VR offers the opportunity to intensify repetitive tasks and increase visual and auditory feedback, making VR therapy more interesting than traditional physical therapy and without posing any serious threat or physical limitations to participants [11]. Previous reviews have examined the effectiveness of VR exercise on physiological, psychological or rehabilitative outcomes. For example, researchers suggested that VR could promote the lower limb function of patients who suffered from stroke [12]. VR exercise has also shown to have a significant effect on the balance ability of patients who suffered from stroke, Parkinson’s disease (PD) or children with cerebral palsy (CP) [13]. Additionally, the effectiveness of the application of VR in psychological treatment in psychotherapy has been widely supported [14]. For instance, VR exercise has been observed to relieve anxiety and depression [15]. As for rehabilitative effectiveness, VR technology in disease rehabilitation has been widely applied, namely to help disabled patients acquire lost motor skills caused by injury or illness and ensure these individuals are able to carry out activities of daily living. As such, the effectiveness of VR exercise on physiological and rehabilitative outcomes have been mostly related and combined.

Thus far, a meta-analysis demonstrated the positive effects of VR exercise on balance function in stroke patients [16]. Similarly, another review suggested that VR was useful for enhancing motor control, functional, and cognitive abilities and balance in Parkinson’s patients [13]. Distinctly, the target populations of the studies within the preceding literature review were narrow and limited to specific diseases, such as strokes and Parkinson’s disease. However, there are other vulnerable populations who may benefit from VR exercise, such as the elderly who could benefit from improved balance and other physical abilities to facilitate better health-related quality of life (HRQoL). However, reviews examining the utilization of VR exercise to intervene in such populations are sparse. Moreover, many relevant reviews are outdated, and thus there is a need to synthesize more updated research. Furthermore, some reviews included single-case experimental designs (i.e., studies with no control group), and thus these reviews were not based on high-quality research and the findings need to be further explored. It has been suggested researchers should include more rigorous study designs like randomized controlled trials (RCTs).

Therefore, this review aimed to fill the existing research gaps. Specifically, most of the articles included in this review were RCTs. Even if some studies could not be randomly grouped on a large scale due to the sample limitations, studies were only included if there were control group(s) and employed a comparative analysis between experimental groups and control groups, as well as between baseline- and post-tests for the examined health outcomes. Further, the target population of this review was relatively extensive, including clinical and healthy populations, allowing the overall effectiveness of VR exercise to be established. Taken together, the purpose of this review is to systematically synthesize the literature examining the effects of VR exercise on the physiological, psychological, and rehabilitative outcomes in various populations.

## 2. Method

The framework and reporting of this review were based on the Preferred Reporting Items for Systematic Review and Meta-Analysis Protocols (PRISMA-P) 2015 statement [17].

### 2.1. Information Sources and Search Strategies

The electronic databases were used to conduct the literature search were as follows: Academic Search Complete, Communication and Mass Media Complete, Education Resources Information Center (ERIC), PubMed, Scopus, Web of Science, PsycINFO, SPORTDiscus, and Medline. Within these databases, the following terms and phrases were searched: (“virtual reality” OR “VR exercise” OR “head-mounted display”) AND (“exercise” OR “physical activity” OR “sports” OR “bike” OR “treadmill”) AND (“physical” OR “movement” OR “physiological outcomes”) AND (“psychology” OR “cognition” OR “mental health” OR “psychological outcomes”) AND (“rehabilitation” OR “recovery” OR “rehabilitative process”). The literature search was conducted independently by all authors and all relevant studies were placed in a shared Google folder.

### 2.2. Eligibility Criteria

Five main eligibility criteria were used to evaluate each study. In detail, articles were included in this review if they: (1) were peer-reviewed and published in English between January 2000 and April 2020; (2) used VR-integrated exercise equipment; (3) intervened on human subjects; (4) used quantitative methods to evaluate the results related to physiological, psychological, and/or rehabilitative outcomes; and (5) employed an established study design (e.g., RCT, controlled trial, and causal-comparative design).

### 2.3. Data Extraction

Three reviewers (authors J.Q., D.M., and Z.G.) screened the titles of potentially relevant articles. The abstracts of these articles were then further reviewed to ensure relevance to the research topic. Data extraction was completed by one reviewer (J.Q.) and checked for accuracy by another reviewer (D.M.). All potential articles were downloaded in full and stored in a shared Google folder and three authors (J.Q., D.M., and Z.G.) reviewed each article independently to ensure that only relevant entries were included. We extracted the following information: (1) publication year and country; (2) study design (i.e., sample characteristics, study duration, VR exposure, results related to physiological, rehabilitation, and/or psychological outcomes, and instruments used); and (3) key findings regarding the effectiveness of VR in outcomes related to the preceding outcomes. Finally, we cross-referenced the bibliographies of the selected articles to further identify the relevant research. It is important to note that no reviewers were influenced by the authors of selected publications or members of journals, nor did we attempt to contact the investigators or journals of the original study for any missing information in the included article.

### 2.4. Risk of Bias in Individual Studies

The process of assessing risk of bias for each study was independently performed by two reviewers (J.Q. and D.M.), using eight quality assessment tools from the previous literature (Table 1) [9,18,19,20]. When the project was clearly described and presented, the study was recorded as “+” (positive) and if the project description was inadequate or missing, the study was recorded as “−” (negative). Further, two reviewers (J.Q. and D.M.) rated each article independently to ensure reliability of the quality assessment. When these two reviewers had different opinions, the third reviewer (Z.G.) re-evaluated the objection. Among all eight indicators, randomization, pre-test/post-test study designs, study retention, and the use a power analysis were considered as the most important factors as they had the most profound impact on research results. The final score for each study was calculated from the sum of all “+” evaluations. The studies which were evaluated as “high quality and low risk of bias” were signified by a score greater than the median score of 5, whereas “low quality and high risk for bias” studies were those which scored lower than the median score 5.

## 3. Results

### 3.1. Study Selection

This study initially included 246 related articles after the initial research. After further examination of the titles and abstracts of these articles, duplicate papers were excluded. Further, by meeting all predetermined eligibility criteria, 15 articles met the inclusion requirements and were included in this review (Figure 1). We chose to disregard some studies upon further investigation for several reasons: (1) articles were not peer-reviewed; (2) articles were not published in English; and (3) articles which did not use VR exercise (e.g., some focused on VR only or VR exposure therapy (VRET) or Virtual Reality in Psychological Treatment (VRT)).

The characteristics of the included studies are shown in the Table 2, which includes 11 RCTs [7,21,22,23,24,25,26,27,30,32,34], three controlled trials [28,31,33] and one causal-comparative study [29]. In detail, all 11 RCTs used baseline and post-test results from the intervention and control groups as the basis of their conclusions [7,21,22,23,24,25,26,27,30,32,34]. Moreover, the three controlled trials compared the inferential statistics at baseline and post-test of the predetermined control and experimental groups [28,31,33]. Lastly, the causal-comparative study analyzed differences between the control group and the experimental group [29].

Experiments were conducted between 2009 and 2019 and were conducted in different countries: six in South Korea [21,24,25,27,32,33], two in Canada [28,30], and one in Brazil [22], Taiwan [23], Israel [26], Australia [29], the US [7], Belgium [31], and Spain [34]. The sample size varied from 11 to 121 and the age of participants ranged from children (five years old) to the elderly (≥70 years old), among which the majority of articles targeted adults and the older populations. The included studies also included specific groups, such as pregnant women [22] and patients with certain diseases (e.g., stroke, cognitive decline) [21,23,24,26,28,30,31,32,34]. Intervention periods varied by study as different studies examined different cause–effect relationships. Studies examining psychological outcomes primarily employed acute interventions (i.e., a single bout of intervention). Further, interventions that examined physiological outcomes employed interventions ranging from two weeks to 12 weeks and were primarily concerned with the effect of VR exercise on the function of the upper and lower limbs, balance and body fitness. The third research query focused on rehabilitation from a disease, such as stroke hemodialysis, spinal-cord, intellectual and development disabilities (IDD), cerebral palsy (CP), and cognitive decline (CD), with the physiological and psychological outcomes mentioned previously. These studies typically had flexible intervention periods, depending on the status of the patient and time of discharge. In these studies, the VR interventions most commonly used were Nintendo products and the specific games played varied based on the different research purposes across studies. For example, Nintendo Balance was used during interventions in which balance was the primary outcome. In terms of study setting, four studies were conducted in a rehabilitation center [21,28,30,31], three in a laboratory setting [7,22,23,29], and the rest did not report specifically where the study was conducted [24,25,26,27,32,33,34].

### 3.2. Quality and Risk of Bias Assessment

The risk assessment table for bias among the included studies is shown in Table 1. In detail, study quality ranged from three points to seven points with a median of five points. Eight of the included studies scored equal to or greater than median score of five and were therefore considered high quality, while six of the included studies scored lower than median score of five and were consequently considered low quality. The majority of the articles retained at least 70% of the participants and the measurement tools in all articles were valid. However, only six articles accounted for the analysis of missing values, five articles employed a power analysis prior to experiment, and no articles reported six-month follow-up post-intervention. The low scores were attributed to missing data, the absence of a power analysis and a lack of follow-up.

### 3.3. Data Items

Among all included studies, the outcomes of interest were divided into three categories: physiology, psychology, and rehabilitation. Noteworthy is the fact that rehabilitation outcomes could not be assessed independently from physiological and/or psychological outcomes which were indicators of the rehabilitation effect. In these articles, the physiological indicators included upper and lower limb function, balance, fitness, body composition, muscle function, and muscular strength. The psychological indicators included tension, depression, affection, attention, and fatigue. Additionally, the diseases of rehabilitation in studies were stoke disease, hemodialysis, spinal cord, intellectual and development disabilities, cerebral palsy on young people and cognitive decline.

### 3.4. Measurement Protocol

The measurement methods in the included studies were valid measurement methods or scales and the process of data collection was carried out by experienced staff. For example, for balance, the Berg Balance Scale (BBS) [25,26,27,28,29,30,31,32] was the most common measurement tool, followed by the Good Balance System and the one-legged stance test [25], the Trunk Control Measurement Scale (TCMS) [31], and the balance testing paradigm used in their previous research [28]. One article used Qualisys Track Manager (QTM) to measure the kinematic variables on sit-to-stand balance in pregnant women [22]. As for limb function and strength, the Wolf Motor Function Test (WMFT) [21,30] was primarily used to assess the function of the upper limbs, followed by the Fugl-Meyer Assessment-Upper Extremity (FMA-UE) [21]. The sit-to-stand test was used to assess function the lower limbs [25], and electromyography was used in the measurement of limb and lower limb muscle activity [33]. As for fitness, two articles on hemodialysis patients measured muscular strength, flexibility, balance, body component, and fatigue by some physical function tests, physical activity questionnaire and HRQoL [24,34], while the other measured heart rate (HR) [26].

As for the psychological outcomes, the Activation-Deactivation Adjective Check List (AD-ACL) was used to assess mood state [7,23], the Physical Activity Affect Scale (PAAS) for affect [29], the Measure of Attentional Focus (MAF) for attention [29], the Geriatric Depression Scale-Korean (GDS-K) and the Korean version of quality of life Alzheimer’s disease (KQOL-AD) scale for HRQoL [32].

### 3.5. The Effectiveness of VR on Physiological Outcomes

A total of 12 studies included the examination of physiological indexes (including eight studies which also targeted rehabilitation), among which eight studies observed VR to have positive effects on certain physiological indexes, including upper and lower limb function, fitness, body composition, balance, muscle function and muscular strength.

All the articles which examined the effectiveness of VR on physiological indicators observed balance ability to be most strongly positively affected, especially among the elderly, specifically the healthy elderly [25,27] and elderly with CD [32]. Further, a positive effect was also found on sitting balance in children with CP [31], while no impact on postural control balance for CP was observed [28]. One more article which examined the balance of sit-to-stand (STS) in the second and third stages of pregnancy also failed to show improvement [22].

In terms of limb function, the included studies observed VR to have a significant impact on lower limb function in elderly people [25] and trunk and lower limb function in young people [33]. Interestingly, however, two studies which examined the function of upper limbs in stroke patients found differential effects [21,30].

Three studies examined individuals’ fitness-related outcomes, but the fitness outcomes were different. Two articles examining balance, flexibility and muscular strength among hemodialysis patients demonstrated a significant impact [24,34]. As for HR on IDD, even if there was a significant improvement statistically, the researchers claimed that it still could not be proven to improve physical fitness [26].

In summary, four studies observed VR to have null effect on physiological outcomes, most notably among pregnant women, those with IDD, patients with CP and patients with chronic stroke.

### 3.6. The Effectiveness of VR on Psychological Outcomes

Among the articles included in this review, there were relatively fewer studies which examined the effect of VR exercise on psychological outcomes. Three of four articles (including one which also targeted rehabilitation) demonstrated positive effects. All of these studies, however, demonstrated VR exercise to effectively relieve fatigue, reduce depression tendency and increase HRQoL, whether the subjects were healthy or suffering from chronic illness [7,23,32]. However, the psychological indexes regarding affection and attention demonstrated null effects [29].

### 3.7. The Effectiveness of VR on Rehabilitative Outcomes

The main difference between the rehabilitation-centered articles and the other included articles was that the participants who were in rehabilitation suffered from physical and/or mental illnesses. Among nine articles, six indicated VR exercise to have positive effects on various rehabilitative outcomes.

In detail, the results showed that VR exercise elicited positive effects on physiological factors, balance [31,32], extremity function [21] and fitness [24], psychological outcomes, and a weakened nervous system from disease [23]. Two studies had a sample of patients with CP, one of which observed null effect on body control [28], while the other demonstrated a positive effect on sitting balance [31]. The results regarding physical fitness showed hemodialysis patients to have positive effects [24,34], while IDD patients observed null effects [26]. In terms of easing tension and depression, both spinal cord and CD patients using VR exercise saw significant reductions [23,32]. Moreover, VR exercise also had a positive effect on the balance of CD patients [32]. As mentioned above, however, the two studies which examined the effect of VR exercise on middle-aged stroke patients’ upper limb function showed different results [21,30].

## 4. Discussion

Exercise has been considered as a prescription for healthy individuals and clinical populations with various diseases [35]. The main purpose of this review was to synthesize and review the latest available literature examining the effects of VR exercise on individuals’ physiological, psychological, and rehabilitation outcomes among various populations (Table 3). Fifteen studies were included in this review, with 12 examining physiological outcomes, four examining psychological outcomes, and nine examining rehabilitation in which the judgment of rehabilitation was determined by physical and/or psychological indicators. Among these studies, the positive effect of VR on health outcomes was greater than 60%. Some studies demonstrated null effects, but no study observed negative effects. Interestingly, some studies observed different findings despite using similar or identical instruments and measurement procedures due to the different study protocols and samples.

With regard to physiological outcomes, the two studies that examined CP in minors (i.e., those >18 years old) [28,31] demonstrated opposing conclusions. Although balance was the main outcome of each study, conflicting tasks and study protocols prompted different results. Indeed, the studies which focused on position control observed null effects [28] while others focusing on sitting balance displayed positive effects [31]. Sitting balance was considered to be relatively easy compared to posture control, and thus short-term effects were more readily observed. Moreover, STS exercises required torque on each joint to complete the task [36] and therefore when going from a seated position to standing specific strategies should be employed to promote proper implementation. For example, when the elderly perform such tasks, support from upper extremities may serve as a beneficial tool by which to increase the stability of the position [22]. Similarly, another study discerning the body control of pregnant women [22] failed to demonstrate a positive effect. Unlike healthy/normal populations during exercise, pregnant women emphasize the safety of their babies rather than trying to enhance their physical fitness or attenuate their chronic diseases, and therefore tend to reduce the range of motion of the exercises [22]. Moreover, it was found that STS movement did not observe any significant differences between pregnant women within different gestation periods, which suggests that pregnant women tend to be aware of their postural instability and in turn this may make them more cautious about falling during functional activities [22].

These findings were also observed concerning the upper limb function of stroke patients, such that conflicting results were observed despite using the same measurement tools and protocols [21,30]. Different instruments and different intervention periods likely contributed to the conflicting results. Functional Electrical Stimulation (FES) prompted beneficial effects in the upper limb functional rehabilitation after a stroke had been reported in a number of clinical trials including the improvement of motor function, movement range, activities of daily living (ADL) and flexibility [37,38,39,40] and was chosen to be a basic treatment of the upper limbs in Lee’s experiment [21]. VR (which used wearable gloves) was used as an aid in the treatment of FES in order to induce persistent movement [21]. However, based on conventional rehabilitation, VR (Nintendo Wii) was applied as an add-on in therapies which aimed to compare against general entertainment, such as bingo [30]. Moreover, the general entertainment was similar to the VR exercise environment so that participants were unable to distinguish between these two studies. Furthermore, these studies employed entirely different intervention periods. Specifically, the study which observed positive effects lasted for four weeks, while the other study only lasted for two weeks. Indeed, such short intervention periods made it difficult to distinguish the true effects of similar add-on therapies. A previous systematic review also reached a similar conclusion that virtual reality technology is more often used as a dose-increasing purpose to enhance the effectiveness of treatment [41]. However, a simple comparison between a comparison system and a commercial gaming system did not reveal significant differences in their effectiveness [41].

In light of psychological outcomes, VR exercise showed positive impacts on relieving mental tension [23,32] and improving HRQoL [7,32]. Traditional physical therapy may be perceived as a repetitive and lifeless task and participants often feel intimidated when confronted with cold, robotic rehabilitation equipment. However, when participants cycled in an interactive virtual environment, they enjoyed cycling, which was deemed as a fun activity rather than a daunting task like traditional therapy. Such improvements in mood and affect during exercise may facilitate greater endurance during exercise and may therefore promote greater effects [23]. VR-based exercise programs, for example, were always seen by patients as games rather than therapeutic situations. Responding to the audio-visual stimuli received through the screen made participants more interested in the program, which facilitated their motor persistence and concentration. Furthermore, patients who were immersed in play-based sports were encouraged by a competitive spirit and motivation to score more points during the game. In these circumstances, the reward circuit of the brain may be activated and dopamine production may increase as a result of the placebo, thereby easing depression [42]. However, little effect was observed for emotion and attention during VR exercises [29]. Young adults, who were accustomed to VR devices found it difficult to concentrate on the exercise and gained satisfaction from it, especially those who were familiar with computer games [7]. On the contrary, the older populations might have more potential to fulfill their curiosity when using novel equipment and may therefore gain pleasure from using equipment like VR during exercise. Gender was also a factor which may have influenced similar conclusions. Generally speaking, male participants tended to play more video games than females, and thus they were more familiar with the VR-enhanced computer experience [42,43] and observed less novelty during experiment. Hence, the experience level of participants with VR apparatuses may have contributed to the observed different results.

In regards to rehabilitative outcomes, the HR results of IDD patients [26] were controversial. Due to the disability status of IDD patients [26], PA is always facilitated by guardians or caretakers, and therefore, this population is heavily sedentary and often has poor health statuses [26]. Additionally, those with IDD lack the ability to participate in sports activities [44], especially those who live in residential environments [45]. But the main reason behind the contradictory results was the choice of fitness-related outcomes. Balance, speed, flexibility, and other outcomes are all indexes of physical fitness. However, these studies used HR as an outcome, which was not a representative fitness indicator.

Overall, VR has potential for enhancing physiological, psychological, and rehabilitative outcomes among healthy and clinical populations, although a few articles observed null effects. To discern the reasons underlying the contradictory results, we must examine the experimental protocols used in the included studies—task, sample, instrument, intervention length, intervention fidelity, experience level with VR, and specific items selected from general measurements were the key to intervention success. We speculate that VR will promote positive effects in the long-term. However, some null effects of immersive devices were found during testing. It was reported that some participants experienced disorientation, nausea and vision problems during or after the use of the VR devices [46].

Since the majority of the included studies were RCTs, the conclusions of this review are relatively valid. However, some limitations of this review must be noted: (1) only articles published in English were included, which may exclude relevant research published in other languages; (2) due to our rigorous inclusion criteria, only a limited number of studies were included in this review; (3) a few included studies had based their protocols in highly-controlled laboratory settings, which may limit the external validity of the findings to some extent, and some studies neglected to report their experimental environment and thus the clinical results need to be further confirmed. Future research should employ high quality designs (e.g., RCTs) among various potential populations (e.g., a depression population) [47]. Additionally, the intervention length of future studies should be extended, and follow-ups should be executed post-intervention to ensure the long-term intervention effectiveness. Moreover, potential confounding factors, such as gender, age, and socio-economic status should also be taken into account. Lastly, different perceptions of VR may produce different effects, and therefore should be discussed carefully in different situations and contexts. Nevertheless, based upon our review of the available literature, we conclude that VR is safe and has potential for improving physiological, psychological, and rehabilitative outcomes if implemented well by trained professionals [48].

The findings of this review have practical implications for researchers and health practitioners. Indeed, utilizing VR-based exercise rehabilitation training in hospitals might be conducive to the physical and mental health of patients due to the fact that VR has positive effects on some aspects of physiology and stress relief during the rehabilitation period. Similar advice may be applied to the public as well, such that using VR properly can be a more life-like and effective way to develop and maintain healthy lifestyles. Given the potential negative effects of immersive VR devices (e.g., motion sickness), rehabilitation centers and healthcare providers can choose other devices, such as non-immersive or interactive devises, based upon reality. Further, the price of VR equipment is inconsistent and must be taken into account. For ordinary consumers, they may choose a light and simple device or use apps on their smartphone which are similar to the VR devices, while the professional rehabilitation centers may choose higher-quality, multi-function VR devices to meet the needs of target populations.

## 5. Conclusions

Our findings suggest that VR exercise has the potential to exert a positive impact on individual’s physiological, psychological, and rehabilitative outcomes compared with traditional exercise. However, the quality, quantity, and sample size of existing studies are far from ideal. Therefore, more rigorous studies are needed to confirm these positive effects.

## Figures and Tables

**Figure 1 ijerph-17-04133-f001:**
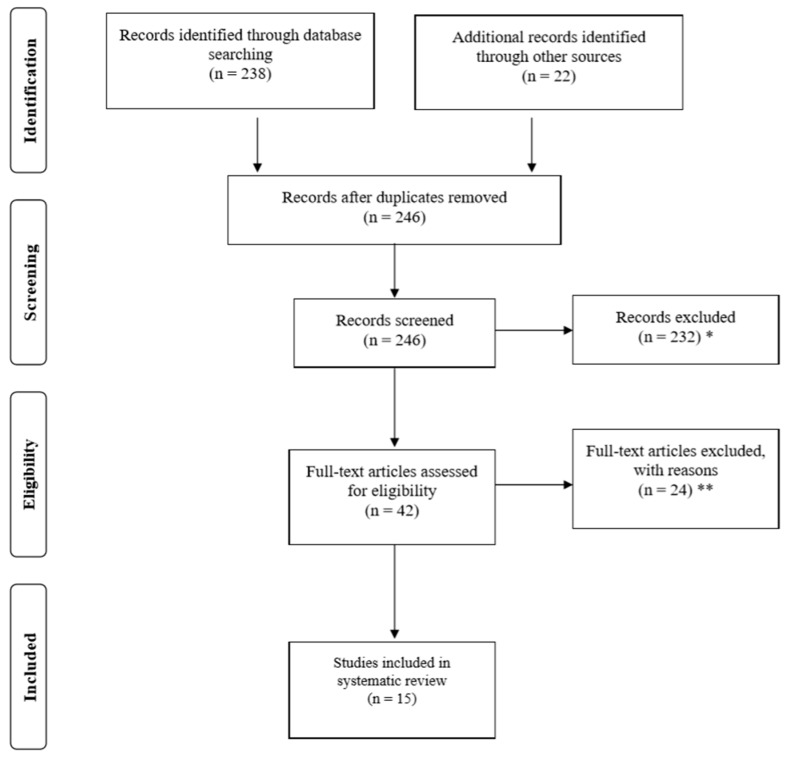
Flow diagram of studies through the review process. Note. * reasons for exclusions included ineligible age, ineligible exposure, ineligible analysis; ** reasons for exclusions included ineligible outcomes and lack of means/standard deviations.

**Table 1 ijerph-17-04133-t001:** Design quality analysis.

Articles	Randomization	Control	Pre–Post	Retention	Mission Data	Power Analysis	Validity Measure	Follow-Up	Score	Effectiveness
Lee et al. [21]	+	+	+	+	+	−	+	−	6	YES
Ribeiro et al. [22]	+	+	+	+	−	−	+	−	5	NA
Chen et al. [23]	+	+	+	−	−	−	+	−	4	YES
Cho et al. [24]	+	+	+	+	+	+	+	−	7	YES
Lee et al. [25]	+	+	+	+	+	+	+	−	7	YES
Lotan et al. [26]	+	+	+	−	−	−	+	−	4	NA
Cho et al. [27]	+	+	+	−	−	−	−	−	3	YES
Mills et al. [28]	−	+	+	−	+	−	+	−	4	NA
Neumann et al. [29]	−	+	−	+	−	+	+	−	4	NA
Saposnik et al. [30]	+	+	+	+	−	+	+	−	6	NA
Plante et al. [7]	+	+	+	−	−	−	+	−	4	YES
Meyns et al. [31]	−	+	+	+	+	+	+	−	6	YES
Lee [32]	+	+	+	+	−	−	+	−	5	YES
Park et al. [33]	−	+	+	−	−	−	+	−	3	YES
Segura-Ortí1 et al. [34]	+	+	+	−	−	+	+	−	5	YES

Note: + refers to positive (explicitly described and present in details); − refers to negative (inadequately described and absent); YES effectiveness indicates significant positive effect; NA indicates no significant effect; Median score = 5. Retention: retaining more than 70% of the participants; Follow-up: following more than 6 months after experiment.

**Table 2 ijerph-17-04133-t002:** Descriptive characteristics of included studies.

Reference	Sample	Testing/Setting	Outcomes/Instrument	Exposure	Dose	Finding
Lee et al. [21] 2018, Korea	RCT; N = 48 (chronic stroke); Intervention (VR + FES) = 20 (49.5 ± 13.7 years); Control (FES) = 21 (46.1 ± 13.0 years).	Baseline, week 2, 4, 8; Stroke rehabilitation.	Upper limb: the Fugl-Meyer Assessment-Upper Extremity (FMA-UE) and Wolf Motor Function Test (WMFT).	The intervention group used a VR-based wearable rehabilitation device and Functional Electrical Stimulation (FES), while the control group used FES only.	Both groups received 20 intervention sessions of 30 min on weekdays (5 times/week) over 4 weeks.	FES with VR-based rehabilitation may be more effective than cyclic FES in improving distal upper extremity gross motor performance poststroke.
Ribeiro et al. [22] 2017, Brazil	RCT; N = 44 (pregnant women); Control Group-2nd Trimester (CG2T) = 10; Experimental Group-2nd Trimester (EG2T) = 8; (2nd Trimester = 28.55 ± 3.83 years); Control Group-3rd Trimester (CG3T) = 10; Experimental Group-3rd Trimester (EG3T) = 16; (3rd Trimester = 29.42 ± 3.23 years).	Pre–post: laboratory.	The sit-to-stand activity kinematic variables: Qualisys Track Manager (QTM).	The intervention group used a Wii balance board (WBB) to exercise additionally, while the control group did not.	An intervention with game therapy was performed in 12 sessions of 30 min each, three times a week in 4 weeks.	The use of the Nintendo Wii Fit Plus was not able to influence sit-to-stand kinematic variables in the pregnant women.
Chen et al. [23] 2009, Taiwan	RCT; N = 30 (incomplete low spinal-cord injuries) (16 women and 14 men, 48.2 ± 18.07 years); Intervention = 15 (51.3 ± 15.83 years); Control = 15 (45.36 ± 14.24 years).	Pre–post; hospital laboratory.	Mood states: the Activation-Deactivation Adjective Check List (AD-ACL).	An experimental group underwent therapy with a VR-based exercise bike, and a control group underwent the therapy without VR equipment.	Acute.	A VR-based rehabilitation program can ease patients’ tension and induce calm.
Cho et al. [24] 2014, Korea	RCT; N = 46 (hemodialysis patients); Exercise = 23 (60.8 ± 6.9 years); Control = 23 (57.7 ± 9.5 years).	Pre–post.	Fitness (strength, flexibility, balance), body composition and fatigue.	While waiting for their dialyses, the exercise group followed a Virtual Reality Exercise Program (VREP), and the control group received only their usual care.	The VREP was accomplished using Nintendo’s Wii Fit Plus for 40 min, 3 times a week for 8 weeks.	The VR Exercise Program improves physical fitness, body composition, and fatigue in hemodialysis patients.
Lee et al. [25] 2017, Korea	RCT; N = 44 (community dwelling older adults); Experimental group = 21 (76.15 ± 4.55 years); Control group = 19 (75.71 ± 4.91 years).	Pre–post.	Static Balance: The Good Balance System and the one leg stance test; dynamic balance: the Berg Balance Scale (BBS); extremity strength: the sit-to-stand test.	The intervention group attended a 60-min VR training session, while the control group did not.	The virtual reality training was conducted for 60 min, twice a week for 6 weeks.	Three-dimensional video gaming technology might be beneficial for improving postural balance and lower-extremity strength in community-dwelling older adults.
Lotan et al. [26] 2010, Israel	RCT; N = 44 (IDD sever level); Experimental group = 20 (47.9 ± 8.6 years); Comparison groups = 24 (46.2 ± 9.3 years).	Pre–post.	Fitness: heart rate (HR).	The intervention groups did game-like exercises provided by a video capture VR system, while the control group did not.	An 8-week fitness program consisting of 2–3 30-min sessions per week.	It is not strong enough functionally to claim that this program improved physical fitness of individuals with severe intellectual disability.
Cho et al. [27] 2014, Korea	RCT; N = 32 (healthy elderly people) VR training group = 17 (73.1 ± 1.1 years); Control group = 15 (71.7 ± 1.2 years).	Pre–post.	Balance: the Romberg test.	The VR training group engaged in an exercise session using Wii Fit, while the control group received no intervention.	A 30-min exercise session using Wii Fit 3 times a week for 8 weeks.	Virtual reality training is effective at improving the balance of the healthy elderly.
Mills et al. [28] 2019, Canada	Control trial; N = 11 (7–17 years) (6 males and 5 females with GMFCS levels I and II); Interactive Rehabilitation Exercise System (IREX) group = 5; Control group = 6.	Pre–post; treatment center.	Balance: the balance testing paradigm.	Participants in the intervention group received 1 h one-on-one physiotherapist-supervised VR balance games for 5 consecutive days between assessments, while the control group received no intervention.	60 min/day in 5 consecutive days.	There was no effect of a 5-day VR-based intervention on postural control mechanisms used in response to oscillating platform perturbations.
Neumann et al. [29] 2018, Australia	Causal-comparative design; N = 40; Virtual Reality group = 24 (Male = 11 Female = 13 M = 24.58 years); Neutral Images group = 16 (Male = 8 Female = 8 M = 24.37 years).	Pre–post.	Affect state: the Physical Activity Affect Scale (PAAS); Attentional States: Measure of Attentional Focus (MAF).	The VR group ran in a computer-generated VR environment that included other virtual runners, while another group ran whilst viewing neutral images.	Depends on 70% VO_2MAX._	VR may not always be better than distracting attention away from exercise-related cues.
Saposnik et al. [30] 2016, Canada	RCT; N = 141 (stroke); VR Wii group = 59 (62 ± 13 years); Recreational activity group = 62 (62 ± 12 years).	Pre–post; rehabilitation center.	Upper extremity motor performance: the Wolf Motor Function Test (WMFT).	The VR Wii group used the Nintendo Wii gaming system to add on conventional rehabilitation, while the control group used simple recreational activities (playing cards, bingo, Jenga, or a ball game).	Ten sessions of 60 min each, over a 2-week period.	Non-immersive virtual reality as an add-on therapy to conventional rehabilitation was not superior to a recreational activity intervention in improving motor function.
Plante et al. [7] 2003, USA	RCT; N = 88 (38.10 ± 12.31 years) Exercise group (E) = 28 VR group (V) = 28 Exercise + VR group (E + V) = 30.	Pre–post; laboratory	Momentary mood states: the Activation-Deactivation Adjective Check List (AD-ACL).	(1) E: bicycling at a moderate intensity (60–70% maximum heart rate) on a stationary bicycle; (2) V: playing a virtual reality computer bicycle game; (3) E + V: an interactive virtual reality bicycle experience on a computer while exercising on a stationary bike at moderate intensity (60–70% maximum heart rate).	30 min.	The combination of virtual reality and exercise might improve some of the beneficial psychological effects of exercise compared with virtual reality or exercise alone.
Meyns et al. [31] 2017, Belgium	Controlled trial; N = 11 (4/7 males/females with spastic CPc following rehabilitation after lower limb orthopedic surgery) (5–18 years); Intervention = 4; Control = 7.	Pre–post; rehabilitation center.	Balance: the Trunk Control Measurement Scale (TCMS).	The control group received conventional physiotherapy, while the intervention group received additional VR training.	The additional VR training was given 3 times a week for 30 min until discharge from the hospital.	Including additional VR training to conventional physiotherapy was feasible and might be promising to train sitting balance in CPc after lower limb surgery.
Lee [32] 2016, Korea	RCT; N = 30 (12 female, 18 male with cognitive decline); Experimental = 15 (63.8 ± 10.2 years); Control groups = 15 (65.5 ± 8.1 years).	Pre–post.	Balance abilities: the Berg Balance Scale (BBS); life quality in patients: Geriatric Depression Scale-Korean (GDS-K) and the Korean version of quality of life Alzheimer’s disease (KQOL-AD) scale.	All subjects performed a traditional cognitive rehabilitation program and the experimental group performed additional VR based video game.	Three 40-min VR based video game (Wii) sessions per week for 12 weeks.	A virtual reality-training program could improve the outcomes in terms of balance, depression, and quality of life in patients with CD.
Park et al. [33] 2014, Korea	Controlled trial; N = 24 (15 males, 9 females); VR exercise group (VREG) = 12 (21.9 ± 1.4 years); Stable surface exercise group (SEG) = 12 (24.3 ± 3.9 years).	Pre–post.	Muscle activities: electromyography.	The VREG used the Nintendo Wii Fit, while the SEG used a stable surface.	Three times a week for six weeks.	Virtual reality exercise using the Nintendo Wii Fit was an effective intervention for the muscle activities of the TA and MG of normal adults.
Segura-Ortí1 et al. [34] 2019, Spain	RCT; N = 40 (hemodialysis patients); CG (control group) = 20; VRG (VR group) = 20.	Pre–post.	Physical activity: the physical function tests, physical activity questionnaire and health-related quality of life (HRQoL).	The VR program used the non-immersive gaming intervention for the VR group, comparing the results to a non-exercising control group.	30 min for 12 weeks.	VR exercise during hemodialysis was safe and improved physical function and HRQoL and could be performed safely toward the end of the hemodialysis session.

Note: RCT = Randomized Controlled Trial; VR = Virtual Reality; FES = Functional Electrical Stimulation; IDD = Intellectual and Developmental Disability; GMFCS = Gross Motor Function Classification System; CPc = Children with Cerebral Palsy; TA = Tibialis Anterior; MG = Medial Gastrocnemius.

**Table 3 ijerph-17-04133-t003:** The main pathologies, function and effect of VR among various populations.

		Main Pathologies	Main Function	Main Effects
Health population		Older	Balance	Induce repetition; enhance motivation; enhance enjoyment.
		Youngers	Muscle activities	
Patient	Physiology	Hemodialysis	Fitness	
		Stroke patient	Limbs strength	
			Balance	
	Psychology	CPc	Relief stress	

Abbreviations: CPc = Children with Cerebral Palsy.

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
