# Peer review of "The Effectiveness of Virtual Reality Exercise on Individual’s Physiological, Psychological and Rehabilitative Outcomes: A Systematic Review"

_ijerph, 2020, doi:10.3390/ijerph17114133_

Round 1
Reviewer 1 Report
A meta-analysis was performed to review virtual reality's (VR) use in the rehabilitation of patients. Specifically, physiological, psychological, and rehabilitative outcomes were reviewed. The authors determined the parameters of the reviewed articles and filtered out the articles that did not meet specific criteria (clinical trial, RCT).
Overall, the authors did well to focus on VR in the assessment of specific clinical measures. Limitations were provided and the authors indicated that additional factors must be considered when assessing VR in the rehabilitative setting.
Author Response
Response to Reviewer 1 Comments
Point 1:
A meta-analysis was performed to review virtual reality's (VR) use in the rehabilitation of patients. Specifically, physiological, psychological, and rehabilitative outcomes were reviewed. The authors determined the parameters of the reviewed articles and filtered out the articles that did not meet specific criteria (clinical trial, RCT).
Overall, the authors did well to focus on VR in the assessment of specific clinical measures. Limitations were provided and the authors indicated that additional factors must be considered when assessing VR in the rehabilitative setting.
Response 1:
We thank the reviewer for their time spent in reviewing our manuscript. Further, we would like to thank the reviewer for their kind comments and for advocating our work. We wish you the best of luck with your future research.
Reviewer 2 Report
This article reviews the impact of VR exercise on different populations. A positive aspect of the manuscript is that is focused on a novel way of exercise that seems to have promising results regarding the increase of motivation and adherence. On the other hand, the first concern is the different nature of pathologies that are reviewed together. I wonder if meta-analysis could be done by grouping the articles regarding outcome measures (physicological, psychological, and rehabilitative variables) and pathologies.
According to the authors, a meta-analysis on VR exercise impact for stroke already exists. I would suggest to further work with the data and meta-analyze results from the RCTs included in your review.
It seems there is a lack of table with excluded criteria showing reasons to exclude them. Please add a table with references and reasons for exclusion in future versions.
Your reference 12 report data from an RCT implementing VR exercise during hemodialysis. Please, add data from this article to your review.
I wonder if rehabilitative and physiological outcomes could be considered within the same category.
I suggest further discussion regarding the type of exercise, was it aerobic (cycling in VR environment)? Was it resistance training? This could be also a criterion to meta-analyze data. I think some clinicians would think VR is not really exercising at some point if we are not able to classify it into one of the traditional exercise options- aerobic vs resistance training.
Please, order legends alphabetically
Summarizing, I think the manuscript has an interesting topic, but I wonder if you have missed other articles, such as your reference 12. I also believe you should meta-analyze data, trying to put together the information in a more quantitative way, and filling the gap of the previous meta-analysis that have not included RCTs as you do.
Author Response
Response to Reviewer 2 Comments
Point 1:
According to the authors, a meta-analysis on VR exercise impact for stroke already exists. I would suggest to further work with the data and meta-analyze results from the RCTs included in your review.
Response 1:
We thank the reviewer for their insight. Obviously, meta-analysis would be ideal as it would quantify overall effects. But higher standards/requirements should be met in meta-analysis, such as similar populations, similar research objects, similar research protocols and similar research instruments across studies. Only in this way, the results of this analysis will be meaningful and reliable/unbiased. However, as noted in our paper, there are relatively few RCTs in this field and the preceding criteria for meta-analysis couldnot be met given the limited available literature in this field of inquiry, rendering it difficult to conduct a meta-analysis. Indeed, VR exercise is relatively new to the field and more studies are needed to compile reliable evidence for future meta-analyses to be conducted.
Point 2:
It seems there is a lack of table with excluded criteria showing reasons to exclude them. Please add a table with references and reasons for exclusion in future versions.
Response 2:
We thank the reviewer for their comments and understand their concerns. Although there is no table regarding the exclusion criteria of our review, this information was already included in the paper and we explained the specific exclusion criteria in section 3.1. (see Page 6) In addition, we also demonstrated this information within the Flow diagram of the article selection process. In the Flow diagram, the steps and criteria of article processing can be found (see Page 6).
Point 3:
Your reference 12 report data from an RCT implementing VR exercise during hemodialysis. Please, add data from this article to your review.
Response 3:
We agree and thank the reviewer for the comment. We have added the data in the review (Page 12, Line 2) (Page 13, Line 13 in 3.5) (Page 15, Line 8 in 3.6).
Point 4:
I wonder if rehabilitative and physiological outcomes could be considered within the same category.
Response 4:
We thank the reviewer for the attention to detail. Indeed, this issue arose during the design phase our study. Upon discussion, we agreed that these two outcomes should not be combined since there are some distinct differences between them.
Regarding physiological outcomes, although some studies had examined certain diseases, they disregarded the healthy populations, and thus we wanted to include this important population. For example, the healthy older population is an important group in our society, but with aging, their balance ability, for example, would be compromised. Whether VR exercise could have some benefits on older adults was one of the goals of our review.
Regarding rehabilitative outcomes, physical rehabilitation is only one aspect. Mental state in the treatment process is also a key indicator of rehabilitation. In the case of COVID-19, for example, isolation has made people physically safe, but now mental illness is on the rise. Therefore, some articles adopted in this review were related to the physiological and psychological factors in the process of disease rehabilitation. That said, we did not want to overlook the psychological outcomes in the process of disease rehabilitation in general, and therefore we chose not to combine the physiological and recovery results together.
Point 5:
I suggest further discussion regarding the type of exercise, was it aerobic (cycling in VR environment)? Was it resistance training? This could be also a criterion to meta-analyze data. I think some clinicians would think VR is not really exercise.
Response 5:
Thank you for this comment. We agree that VR is not always exercise but the scope of this review was to examine VR-integrated exercise. Indeed, an important criterion in the selection of the included articles was that they had to include exercise when using VR. On the other end, in the field of psychology, VR has been used in some therapies without doing exercise. Therefore, when selecting articles, we focused more on this issue.
Secondly, because there are so few articles at present in this specific field of inquiry and many aspects to evaluate, such as balance, upper body strength, among other specific items, such as STS balance and standing balance, there is no current unified standard by which to conduct a meta-analysis. As more research compiles, we plan to conduct a meta-analysis in the future.
Lastly, the particularity of the population, most of whom were sick or elderly, means their exercise intensity could not reach that of healthy populations, so we found it relatively difficult to identify exercises aerobic or resistance exercise.
Point 6:
Please, order legends alphabetically
Response 6:
Thank you for your notification, we have revised it (Page 16).
Reviewer 3 Report
Very interesting article that summarizes 30 years of using Virtual Reality
I have only two suggestions:
1. Why not include other systematic reviews in the references and discussion?
Laver, K. E., Lange, B., George, S., Deutsch, J. E., Saposnik, G., & Crotty, M. (2017). Virtual reality for stroke rehabilitation. The Cochrane Database of Systematic Reviews, 2, CD008349.
2. It would be interesting for the reader to produce a table presenting the main pathologies in which VR has been used:
Stroke
Parkinson
Multiple sclerosis
Cerebral Palsy...
and summarize the main effects according to the pathology:
Enhance motivation/repetition
Provide precise feedback
Transferability
Early stage learning...
Author Response
Response to Reviewer 3 Comments
Point 1:
Why not include other systematic reviews in the references and discussion?
Response 1:
We understand the reviewer’s concern for not referencing other systematic reviews in our discussion. However, the major scope of this review was namely to explore the influence of VR exercise on the general population or clinical populations. We observed that, as a whole, VR exercise had a positive impact. However, some interesting findings emerged such that similar research methods have been conducted like similar surveys employed to similar samples yet produced different results. Therefore, in the discussion section, we focused on those “interesting findings” in detail as this would facilitate future research to try and fill these knowledge gaps. Indeed, there were some other somewhat similar review studies but they were beyond the scope of our study and therefore were not included.
However, for the literature you proposed, we found more meaningful aspects, so they were added to a certain part of the discussion (Page 16, Line1-3).
Point 2:
It would be interesting for the reader to produce a table presenting the main pathologies in which VR has been used:
Response 2:
We thank the reviewer for your thoughtful insight. We agree that providing a table to summarize the results of the study will make the conclusions clearer to readers. That said, we have added a new table to the results/discussion section, which demonstrated the main pathologies, function and effect of VR on populations included in the review (Page15).
Table 3. the Main Pathologies, Function and Effect of VR among Various Populations.
|
Main pathologies |
Main function |
Main effects |
|
Health Population |
Older |
Balance |
Induce repetition Enhance motivation Enhance enjoyment |
|
Youngers |
Muscle activities |
|||
Patient |
Physiology |
Hemodialysis |
Fitness |
|
Stroke patient |
Limb strength |
|||
Balance |
||||
Psychology |
CPc |
Relief stress |
Abbreviations: CPc, Children with Cerebral Palsy.